# PortaSpeech: Portable and High-Quality Generative Text-to-Speech

**Yi Ren**[*]
Zhejiang University
rayeren@zju.edu.cn

**Jinglin Liu**[*]
Zhejiang University
jinglinliu@zju.edu.cn

**Zhou Zhao**[†]
Zhejiang University
zhaozhou@zju.edu.cn

## Abstract

Non-autoregressive text-to-speech (NAR-TTS) models such as FastSpeech 2 [24] and Glow-TTS [8] can synthesize high-quality speech from the given text in parallel. After analyzing two kinds of generative NAR-TTS models (VAE and normalizing flow), we find that: VAE is good at capturing the long-range semantics features (e.g., prosody) even with small model size but suffers from blurry and unnatural results; and normalizing flow is good at reconstructing the frequency bin-wise details but performs poorly when the number of model parameters is limited. Inspired by these observations, to generate diverse speech with natural details and rich prosody using a lightweight architecture, we propose PortaSpeech, a portable and high-quality generative text-to-speech model. Specifically, 1) to model both the prosody and mel-spectrogram details accurately, we adopt a lightweight VAE with an enhanced prior followed by a flow-based post-net with strong conditional inputs as the main architecture. 2) To further compress the model size and memory footprint, we introduce the grouped parameter sharing mechanism to the affine coupling layers in the post-net. 3) To improve the expressiveness of synthesized speech and reduce the dependency on accurate fine-grained alignment between text and speech, we propose a linguistic encoder with mixture alignment combining hard word-level alignment and soft phoneme-level alignment, which explicitly extracts word-level semantic information. Experimental results show that PortaSpeech outperforms other TTS models in both voice quality and prosody modeling in terms of subjective and objective evaluation metrics, and shows only a slight performance degradation when reducing the model parameters to 6.7M (about 4x model size and 3x runtime memory compression ratio compared with FastSpeech 2). Our extensive ablation studies demonstrate that each design in PortaSpeech is effective[3].

## 1 Introduction

Recently, deep learning-based text-to-speech (TTS) has attracted a lot of attention in speech community [2, 15, 20, 22, 24, 25, 29, 35]. Among neural network-based TTS systems, some of them generate mel-spectrograms autoregressively from text [15, 22, 29, 35] and suffer from slow inference speed and robustness (word skipping and repeating) problems [25], while others [8, 12, 16, 19, 24, 25] generate mel-spectrograms in parallel with comparable quality using non-autoregressive architecture, called NAR-TTS, which enjoys fast inference and avoids robustness issues in the meanwhile. In general, modern TTS models aim to achieve the following goals:

- *Fast*: to reduce the cost of computational resources and apply the model to real-time applications, the inference speed of TTS model should be fast.

---

[*]Equal contribution.

[†]Corresponding author

[3]Audio samples are available at https://portaspeech.github.io/.

35th Conference on Neural Information Processing Systems (NeurIPS 2021).

- *Lightweight*: to deploy the model to mobile or edge devices, the model size should be small and the runtime memory footprint should be low.
- *High-quality*: to improve the naturalness of synthesized speech, the model should capture the details (frequency bins between two adjacent harmonics, unvoiced frames and high-frequency parts) in natural speech.
- *Expressive*: to generate expressive and dynamic speech, the model should use powerful prosody modeling methods to accurately model the fundamental frequency and duration of speech.
- *Diverse*: to prevent the synthesized speech from being too dull and tedious when generating long speech, the model should be able to generate diverse speech samples with different intonations given one text input sequence.

To achieve the above goals, in this work, we propose PortaSpeech, a portable and high-quality generative text-to-speech model, which generates mel-spectrograms with natural details and expressive prosody using a lightweight architecture. Specifically,

- Through some preliminary experiments (see Section 4.2), we find that VAE is good at capturing the long-range semantics features (*e.g.*, prosody), while normalizing flow is good at reconstructing the frequency bin-wise details. Based on these observations, we adopt VAE with an enhanced prior followed by a flow-based post-net as the main model architecture of PortaSpeech, which helps PortaSpeech generate *high-quality* and *expressive* results. In addition, PortaSpeech can generate *diverse* speech by sampling latent variables from the prior of VAE and post-net.
- Through the experiments, we also find that even when the model is very small, VAE is still good at capturing the prosody, making it possible for PortaSpeech to reduce its model size using a lightweight VAE. Besides, we introduce the grouped parameter sharing mechanism to the post-net to compress its model size. By doing these, PortaSpeech can be very *lightweight* and *fast* at a small performance cost.
- To model the prosody better and generate more *expressive* speech, we introduce a linguistic encoder with mixture alignment, which combines hard word-level alignment and soft phoneme-level alignment. Our proposed linguistic encoder also reduces the dependence on fine-grained (phoneme-level) alignment and alleviates the burden of the speech-to-text aligner.

Experiments on the LJSpeech [7] dataset show that PortaSpeech outperforms other state-of-the-art TTS models with comparable model parameters in voice quality and prosody in terms of both subjective and objective evaluation metrics. When compressing the model size, our PortaSpeech shows only a slight performance degradation but enjoys the benefits of a much smaller number of model parameters (about 4x model size reduction) and lower memory footprints (about 3x memory reduction) compared with FastSpeech 2. The main contributions of this work are summarized as follows:

- We analyze the characteristics of VAE and normalizing flow when applied to TTS and combines the advantages of VAE and normalizing flow to generate mel-spectrograms with rich details and expressive prosody.
- We propose mixture alignment in the linguistic encoder, which improves the prosody and reduces the dependence on fine-grained (phoneme-level) hard alignment.
- Using lightweight VAE and introducing the grouped parameter sharing mechanism to the post-net, PortaSpeech can generate high-quality speech with a small number of model parameters and small runtime memory footprints.

## 2   Background

In this section, we describe the background of TTS and the basic knowledge of VAE and normalizing flow. We also review the existing applications of VAE and normalizing flow in non-autoregressive TTS and analyze their advantages and disadvantages.

**Text-to-Speech**   Text-to-speech (TTS) models convert input text or phoneme sequence into mel-spectrogram (*e.g.*, Tacotron [35], FastSpeech [25]), which is then transformed to waveform using vocoder (*e.g.*, WaveNet [33]), or directly generate waveform from text (*e.g.*, FastSpeech 2s [24]

and EATS [5]). End-to-end text-to-speech models have gradually developed from autoregressive to non-autoregressive architecture: early autoregressive text-to-speech models [29, 35] generate each mel-spectrogram frame conditioned on previous ones, resulting in high inference latency and low robustness. Recently, several non-autoregressive TTS works have been proposed, which generate mel-spectrogram frames in parallel. FastSpeech [25] and ParaNet [21] are the first non-autoregressive TTS models, which use pre-trained autoregressive TTS teacher models to extract text-to-spectrogram alignments from the training data to bridge the length gap between text and speech for non-autoregressive student model. FastSpeech 2 [24] introduces more variation information of speech, including pitch and energy, to alleviate the one-to-many mapping problem in TTS. While these methods need external text-to-spectrogram alignment models or tools, Glow-TTS [8] directly searches for the most probable monotonic alignment between text and the latent representation of speech using normalizing flows and dynamic programming. In addition to improving the performance of non-autoregressive models, some works focus on lightweight and portable model designs: SpeedySpeech [32] replaces the self-attention layers with fully convolutional blocks to reduce the computational complexity. LightSpeech [17] leverages neural architecture search (NAS) to automatically design more lightweight models, while the training of NAS consumes huge resources. In this work, we save the model parameters by taking advantage of the characteristics of VAE and normalizing flow and introducing the grouped parameter sharing mechanism.

**VAE** The VAE is a generative model in the form of $p_\theta(\mathbf{x}, \mathbf{z}) = p(\mathbf{z})p_\theta(\mathbf{x}|\mathbf{z})$, where $p(\mathbf{z})$ is a prior distribution over latent variables $\mathbf{z}$ and $p_\theta(\mathbf{x}|\mathbf{z})$ is the likelihood function that generates data $x$ given latent variables $\mathbf{z}$ which can be considered as a decoder. It is parameterized by a neural network $\theta$. Since the true posterior $p_\theta(\mathbf{x}, \mathbf{z})$ over the latent variables of a VAE is usually analytically intractable, we approximate it with a variational distribution $q_\phi(\mathbf{z}|\mathbf{x})$, which can be viewed as an encoder. The parameters $\theta$ and $\phi$ can be optimized by maximizing the *evidence lower bound* (ELBO):

$$\log p_\theta(\mathbf{x}) \geq \mathbb{E}_{q_\phi(\mathbf{z}|\mathbf{x})} \left[ \log \frac{p_\theta(\mathbf{x}, \mathbf{z})}{q_\phi(\mathbf{z}|\mathbf{x})} \right] = E_{\mathbf{z} \sim q_\phi(\mathbf{z}|\mathbf{x})} \left[ \log p_\theta(\mathbf{x}|\mathbf{z}) - \log \frac{q_\phi(\mathbf{z}|\mathbf{x})}{p_\theta(\mathbf{z})} \right]$$

$$= E_{\mathbf{z} \sim q_\phi(\mathbf{z}|\mathbf{x})} \left[ \log p_\theta(\mathbf{x}|\mathbf{z}) \right] - \mathrm{KL} \left( q_\phi(\mathbf{z}|\mathbf{x}) \| p_\theta(\mathbf{z}) \right) \equiv \mathcal{L}(\theta, \phi).$$

Recently, some works successfully apply VAE to TTS. One of them is BVAE-TTS [14], which adopts a bidirectional-inference variational autoencoder that learns hierarchical latent representations using both bottom-up and top-down paths to increase its expressiveness. Thanks to the hierarchical structure and latent modeling, BVAE-TTS can capture the dynamism and variability of ground-truth prosody. However, its generated mel-spectrograms are very blurry and over-smoothing, resulting in unnatural sounds, due to the posterior collapse [1, 6] and the reconstruction loss term used in BVAE-TTS, which has independency assumption of generated frequency bins given latent variables.

**Normalizing Flow** Normalizing flow is a kind of generative models [3, 4] which has several advantages including exact log-likelihood evaluation and fully-parallel sampling. In generation, normalizing flows [3, 4] transform the latent variable $\mathbf{z}$ into a datapoint $\mathbf{x}$ through a composition of invertible functions $\mathbf{f} = \mathbf{f}_1 \circ \mathbf{f}_2 \circ \cdots \circ \mathbf{f}_K$ and we assume a tractable prior $p_{\boldsymbol{\theta}}(\mathbf{z})$ over latent variable $\mathbf{z}$ sampled from a simple distribution (*e.g.*, a Gaussian distribution). In training, the log-likelihood of a datapoint $\mathbf{x}$ can be computed exactly using the change of variables rule:

$$\log p_{\boldsymbol{\theta}}(\mathbf{x}) = \log p_{\boldsymbol{\theta}}(\mathbf{z}) + \sum_{i=1}^{K} \log |\det(d\mathbf{h}_i/d\mathbf{h}_{i-1})|, \tag{1}$$

where $\mathbf{h}_0 = \mathbf{x}$, $\mathbf{h}_i = \mathbf{f}_i(\mathbf{h}_{i-1})$, $\mathbf{h}_K = \mathbf{z}$ and $|\det(d\mathbf{h}_i/d\mathbf{h}_{i-1})|$ is the Jacobian determinant. We learn the parameters of $\mathbf{f}_1 \ldots \mathbf{f}_K$ by maximizing Equation (1) over the training data. Given $\mathbf{g} = \mathbf{f}^{-1}$, we can now generate a sample $\hat{\mathbf{x}}$ by sampling $\mathbf{z} \sim p_{\boldsymbol{\theta}}(\mathbf{z})$ and computing $\hat{\mathbf{x}} = \mathbf{g}(\mathbf{z})$.

There are several normalizing flow-based non-autoregressive TTS methods: Flow-TTS [19] is an early flow-based TTS method, which replaces the decoder in FastSpeech with Glow [9] and jointly learns the alignment and mel-spectrogram generation through a single network. Then Glow-TTS [8] is proposed, which combines the normalizing flow and dynamic programming-based monotonic alignment to enable fast, diverse and controllable speech synthesis. These methods handle the blurry mel-spectrogram problems well due to the nature of the normalizing flow. However, according to our experiments (see Section 4.2), flow-based NAR-TTS model usually requires a huge model capacity to achieve good performance, and the performance can drop notably when reducing the number of model parameters.

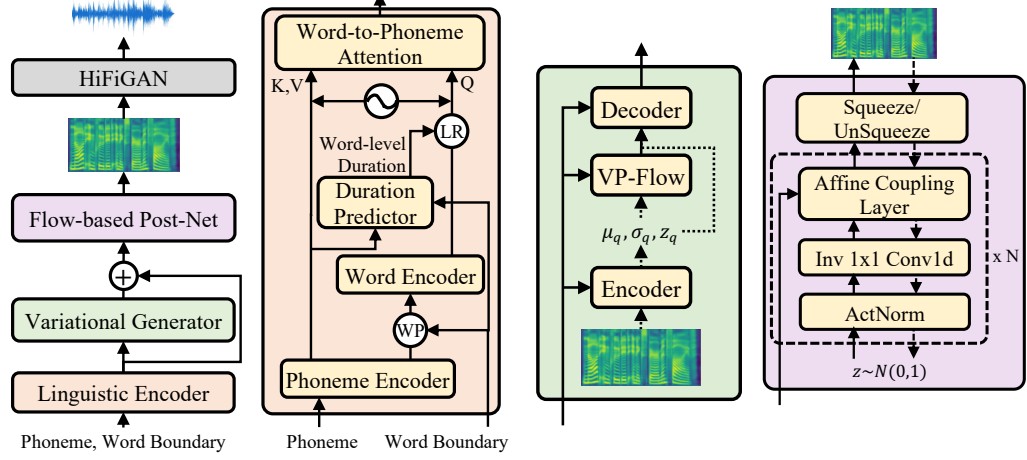

(a) PortaSpeech    (b) Linguistic Encoder    (c) Variational Generator    (d) Flow-based Post-Net

Figure 1: The overall architecture for PortaSpeech. In subfigure (b), "WP" denotes the word-level pooling operation, "LR" denotes the length regulator proposed in FastSpeech and "sinusoidal-like symbol" denotes the positional encoding. In subfigure (c), "VP-Flow" denotes the volume-preserving normalizing flow. In subfigure (c) and (d), the operations denoted with dotted lines are only used in the training procedure.

## 3    PortaSpeech

Considering the characteristics of VAE and normalizing flow mentioned in Section 2, to build a TTS system that can meet the goals described in Section 1, we propose PortaSpeech, which combines the advantages of VAE and normalizing flows and overcomes their deficiencies. As shown in Figure 1a, PortaSpeech is composed of a linguistic encoder with mixture alignment, a variational generator with enhanced prior and a flow-based post-net with the grouped parameter sharing mechanism. First, the text sequence with word-level boundary is fed into the linguistic encoder to extract the linguistic features in both phoneme and word level. Secondly, to model the expressiveness and variability of speech with lightweight architecture, we train the VAE-based variational generator to maximize the ELBO over the ground-truth mel-spectrograms conditioned on the linguistic features, whose prior distribution is modeled by a small volume-preserving normalizing flow. Finally, to refine and enhance the natural speech details in the generated mel-spectrograms, we train the post-net by maximizing the likelihood of ground-truth mel-spectrograms conditioned on both the linguistic features and the outputs of the variational generator. During inference, the text is transformed to mel-spectrograms by successively passing through the linguistic encoder, the decoder of the variational generator and the reversed flow-based post-net. We describe these designs and the training and inference procedures in detail in the following subsections. We put more details in Appendix A.

### 3.1    Linguistic Encoder with Mixture Alignment

To expand the lengths of linguistic features (outputs of the linguistic encoder), previous non-autoregressive TTS models introduce a duration predictor to predict the number of frames of each phoneme (phoneme duration) and the ground-truth phoneme duration (hard alignment) is obtained by external models/tools (*e.g.*, FastSpeech [25] and FastSpeech 2 [24]) or jointly monotonic alignment training (*e.g.*, Glow-TTS [8] and BVAE-TTS [14]). However, phoneme-level hard alignment has several issues: since some of the boundaries between two phonemes are naturally uncertain[4], it is challenging for the alignment model to obtain very accurate phoneme-level boundaries, which inevitably introduces errors and noises. Further, these alignment errors and noises can affect the training of duration predictor, which hurts the prosody of the generated speech in inference. To tackle

---

[4]It could be difficult to determine the exact boundary between two phonemes in millisecond level even for manually labeling.

these problems, we introduce mixture alignment to the linguistic encoder, which uses soft alignment in phoneme level and keeps hard alignment in word level.

As shown in Figure 1b, our linguistic encoder consists of a phoneme encoder, a word encoder, a duration predictor and a word-to-phoneme attention module and detailed architecture of these modules are put in Appendix A.1. Suppose we have an input phoneme sequence together with the word boundary (for example, "HH AE1 Z | N EH1 V ER0", where "|" denotes the word boundary in phoneme sequence). First, we encode the phoneme sequence into phoneme hidden states $\mathcal{H}_p$. Then we apply word-level pooling on $\mathcal{H}_p$ to obtain the input representation of the word encoder, which averages the phoneme hidden states inside each word according to the word boundary. The word encoder then encodes the word-level hidden states into word-level hidden states and expanded them to match the length of the target mel-spectrogram (denoted as $\mathcal{H}_w$) using length regulator with the word-level duration. Finally, to add fine-grained linguistic information, we introduce a word-to-phoneme attention module, which takes $\mathcal{H}_w$ as the query and $\mathcal{H}_p$ as the key and the value. In addition, due to the monotonic nature of text-to-spectrogram alignment, to encourage the attention to be close to the diagonal, we add a word-level relative positional encoding embedding to both $\mathcal{H}_p$ and $\mathcal{H}_w$ before they are fed into the attention module. To predict the word-level duration, we use the duration predictor which takes $\mathcal{H}_p$ as input and then sums the predicted duration of the phonemes in each word as the word-level duration[5]. Our mixture alignment mechanism avoids the uncertain and noisy phoneme-level alignment extraction and duration prediction while keeping fine-grained, soft and close-to-diagonal text-to-spectrogram alignment.

### 3.2 Variational Generator with Enhanced Prior

To achieve *expressive* and *diverse* speech generation with *lightweight* architecture, we introduce VAE as the mel-spectrogram generator, called variational generator. However, traditional VAE uses simple distribution (*e.g.*, Gaussian distribution) as the prior, which results in strong constraints on the posterior: optimizing with Gaussian prior pushes the posterior distribution towards the mean, limiting diversity and hurting the generative power [18, 31]. To enhance the prior distribution, inspired by [10, 18, 26, 27], we introduce a small volume-preserving normalizing flow[6], which transforms simple distributions (*e.g.*, Gaussian distribution) to complex distributions through a series of K invertible mappings (a stack of WaveNet residual blocks with dilation 1). Then we take the complex distributions as the prior of the VAE. When introducing normalizing flow-based enhanced prior, the optimization objective of the mel-spectrogram generator becomes:

$$\log p(\mathbf{x}|c) \geq \mathbb{E}_{q_\phi(\mathbf{z}|\mathbf{x},c)}[\log p_\theta(\mathbf{x}|\mathbf{z},c)] - \mathrm{KL}(q_\phi(\mathbf{z}|\mathbf{x},c)|p_{\bar{\theta}}(\mathbf{z}|c)) \equiv \mathcal{L}(\phi,\theta,\bar{\theta}), \qquad (2)$$

where $\phi$, $\theta$ and $\bar{\theta}$ denote the model parameters of VAE encoder, VAE decoder and the normalizing flow-based enhanced prior, respectively; $c$ denotes the outputs of linguistic encoder. Due to the introduction of normalizing flows, the KL term in Equation (2) no longer offers a simple closed-form solution. So we estimate the expectation w.r.t. $q_\phi(\mathbf{z}|\mathbf{x},c)$ via Monte-Carlo method by modifying the KL term:

$$\mathrm{KL}(q_\phi(\mathbf{z}|\mathbf{x},c)|p_{\bar{\theta}}(\mathbf{z}|c)) = \mathbb{E}_{q_\phi(\mathbf{z}|\mathbf{x},c)}[\log q_\phi(\mathbf{z}|\mathbf{x},c) - \log p_{\bar{\theta}}(\mathbf{z}|c)]. \qquad (3)$$

As shown in Figure 1c, in training, the posterior distribution $N(\mu_q, \sigma_q)$ is encoded by the encoder of the variational generator. Then $z_q$ is sampled from the posterior distribution using reparameterization and is passed to the decoder of the variational generator (the right dotted line). In the meanwhile, the posterior distribution is fed into the VP-Flow to convert it to a standard normal distribution (the middle dotted line). In inference, VP-Flow converts a sample in the standard normal distribution into a sample $z_p$ in the prior distribution of the variational generator and we pass the $z_p$ to the decoder of the variational generator.

### 3.3 Flow-based Post-Net

To generate *high-quality* mel-spectrograms, normalizing flows [8, 19] have been widely proved to be effective. Unlike simple loss-based (L1 or MSE-based) or VAE-based methods that often generate

---

[5]In training, the ground-truth word-level duration can be obtained by external forced alignment tools or autoregressive TTS models.

[6]For simplicity and convenience, we use volume-preserving flow (VP-Flow), which does not need to consider the Jacobian term when calculating the data log-likelihood. We find that volume-preserving is powerful enough for modeling the prior.

blurry outputs, flow-based models can overcome the over-smoothing problem and generate more realistic outputs. To model rich details in ground-truth mel-spectrograms, we introduce a flow-based post-net with strong condition inputs to refine the outputs of the variational generator. As shown in Figure 1d, the architecture of the post-net adopts Glow [9] and is conditioned on the outputs of the variational generator and the linguistic encoder. In training, the post-net transforms the mel-spectrogram samples into latent prior distribution (isotropic multivariate Gaussian) and calculates the exact log-likelihood of the data using the change of variables. In inference, we sample the latent variables from the latent prior distribution and pass them into the post-net reversely to generate the high-quality mel-spectrogram.

However, flow-based models suffer from large model footprints. Since the conditional inputs contain the text and prosody information, our post-net only focuses on modeling the details in mel-spectrograms, greatly reducing requirements for model capacity. To further reduce the model size and keep the modeling power, we introduce the grouped parameter sharing mechanism to the affine coupling layer, which shares some model parameters among different flow steps ($\mathbf{f}_i$, $\mathbf{f}_{i+1}$, ..., $\mathbf{f}_j$). As shown in Figure 2, we divide all flow steps ($\mathbf{f}_1$, $\mathbf{f}_2$, ..., $\mathbf{f}_K$) into several groups and share the model parameters of $NN$ (WaveNet-like network, see Appendix A.3) in the coupling layers among flow steps in a group. Our grouped parameter sharing mechanism is similar to the shared neural density estimator proposed in [13] with some differences that: 1) we simplify the model by removing the flow indication embedding since the unshared conditional projection layer in different flow steps can help the model to indicate the position of the step; 2) instead of sharing the parameters among all flow steps, we generalize the sharing mechanism by sharing the parameters among flow steps in a group, making it easier to adjust the number of trainable model parameters without changing the model architecture.

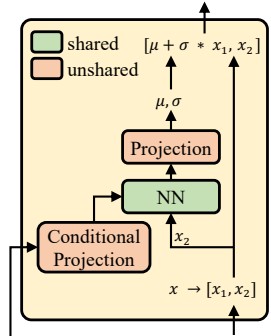

Figure 2: Affine coupling layer with grouped parameter sharing. Green block means sharing the model parameters of this block among flow layers in a group.

### 3.4 Training and Inference

In training, the final loss of PortaSpeech consists of the following loss terms: 1) duration prediction loss $L_{dur}$: MSE between the predicted and the ground-truth word-level duration in log scale; 2) reconstruction loss of variational generator $L_{VG}$: MAE between the ground-truth mel-spectogram and that generated by the variational generator; 3) the KL-divergence of variational generator $L_{KL}$ = $\log q_\phi(\mathbf{z}|\mathbf{x}, c) - \log p_{\bar{\theta}}(\mathbf{z}|c)$, where $\mathbf{z} \sim q_\phi(\mathbf{z}|\mathbf{x}, c)$, according to Equation (3); and 4) the negative log-likelihood of the post-net $L_{PN}$. In inference, the linguistic encoder first encodes the text sequence, predicts the word-level duration and expand the hidden states via mixture alignment to obtain the linguistic hidden states $\mathcal{H}_L$. Secondly, we sample $\mathbf{z}$ from the enhanced prior, and then the decoder of the variational generator generates the coarse-grained mel-spectrograms $\bar{M}_c$ (the output mel-spectrograms before post-net) conditioned on the linguistic hidden states $\mathcal{H}_L$. Thirdly, the post-net converts randomly sampled latent into fine-grained mel-spectrograms $\bar{M}_f$ conditioned on $\mathcal{H}_L$ and $\bar{M}_c$. Finally, $\bar{M}_f$ is transformed to waveform using a pre-trained vocoder. Since we use hard word-level alignment in PortaSpeech, absolute durations for individual words can also be specified at inference time like FastSpeech [25]. As for silences, we add a word boundary symbol as an extra special word such as "SIL" between two words in training. In this way, we can adjust the duration of silences via modifying the duration of the special word "SIL".

## 4 Experiments

### 4.1 Experimental Setup

**Datasets** We evaluate PortaSpeech on LJSpeech dataset [7], which contains 13100 English audio clips and corresponding text transcripts. Following FastSpeech 2 [24], we split LJSpeech dataset into three subsets: 12229 samples for training, 348 samples (with document title LJ003) for validation and 523 samples (with document title LJ001 and LJ002) for testing. We randomly choose 50 samples in the test set for subjective evaluation and use all testing samples for objective evaluation. We convert the text sequence to the phoneme sequence [2, 25, 29, 30, 35] with an open-source grapheme-to-

phoneme tool[7]. We transform the raw waveform with the sampling rate 22050 into mel-spectrograms following [25, 29] with the frame size 1024 and the hop size 256.

**Model Configuration**    Our PortaSpeech consists of an encoder, a variational generator and a post-net. The encoder consists of multiple feed-forward Transformer blocks [25] with relative position encoding [28] following Glow-TTS [8]. The encoder and decoder in variational generator are 2D-convolution networks. The post-net adopts the architecture of Glow [9]. We conduct experiments on two settings with different model sizes: *PortaSpeech (normal)* and *PortaSpeech (small)*. We add more detailed model configurations of these two settings in Appendix B.

**Training and Evaluation**    We train the PortaSpeech on 1 NVIDIA 2080Ti GPU, with batch size of 64 sentences on each GPU. We use the Adam optimizer with $\beta_1 = 0.9$, $\beta_2 = 0.98$, $\varepsilon = 10^{-9}$ and follow the same learning rate schedule in [34]. It takes 320k steps for training until convergence. The output mel-spectrograms of our model are transformed into audio samples using HiFi-GAN [11][8] trained in advance. We conduct the MOS (mean opinion score) and CMOS (comparative mean opinion score) evaluation on the test set to measure the audio quality via Amazon Mechanical Turk. We keep the text content consistent among different models to exclude other interference factors, only examining the audio quality or prosody. Each audio is listened by at least 20 testers. We analyze the MOS and CMOS in two aspects: prosody (naturalness of pitch, energy and duration) and audio quality (clarity, high-frequency and original timbre reconstruction), and score MOS-P/CMOS-P and MOS-Q/CMOS-Q corresponding to the MOS/CMOS of prosody and audio quality. We tell the tester to focus on one aspect and ignore the other aspect when scoring MOS/CMOS of this aspect. We put more information about the subjective evaluation in Appendix B.2.

## 4.2    Preliminary Analyses on VAE and Flow

In image generation tasks, VAE is good at capturing the overall image structure information (low-frequency parts) while discarding small sharp textures/details (high-frequency parts). Similarly, in mel-spectrograms, low-frequency parts correspond to the shape of harmonics, which determines the pitch and prosody of speech. Thus we can intuitively infer that VAE is good at modeling the prosody while not good at modeling the details in speech. While flow-based models can generate high-quality images at the cost of very large model size and huge computation complexity and we may infer that flow-based models can model the details in speech well with large model size.

Table 1: The audio performance comparisons among different NAR-TTS models with different numbers of model parameters (#Params.). GT (voc.) denotes the waveform reconstructed from ground-truth mel-spectrograms using HiFi-GAN [11].

| Methods | Configs | MOS-P | MOS-Q | #Params |
|---|---|---|---|---|
| *GT (voc.)* | / | $4.49 \pm 0.07$ | $4.16 \pm 0.06$ | / |
| *Flow-based* | big | $3.71 \pm 0.06$ | $\mathbf{3.96 \pm 0.07}$ | 41.2M |
|  | middle | $3.52 \pm 0.07$ | $3.54 \pm 0.12$ | 10.2M |
|  | small | $3.21 \pm 0.12$ | $3.42 \pm 0.14$ | 4.5M |
| *VAE-based* | big | $\mathbf{3.81 \pm 0.07}$ | $3.75 \pm 0.08$ | 43.2M |
|  | middle | $3.79 \pm 0.08$ | $3.69 \pm 0.09$ | 9.3M |
|  | small | $3.72 \pm 0.08$ | $3.51 \pm 0.11$ | 4.4M |

To verify our hypothesis and explore the characteristic of VAE and flow-based models in TTS, we conduct audio quality (MOS-Q) and prosody (MOS-P) comparisons among several VAE and flow-based NAR-TTS models with different model sizes: 1) *big*: more than 40M model parameters; 2) *middle*: about 10M model parameters; and 3) *small*: about 5M model parameters. We keep the architecture of the encoders in three models consistent. The detailed model architecture and configurations are put in Appendix A.4. The results are shown in Table 1. From the table, we can see that 1) when reducing the model capacities, the prosody quality of flow-based models drops significantly. In contrast, that of VAE-based model only drops slightly, according to MOS-P. This

---

[7]https://github.com/Kyubyong/g2p
[8]https://github.com/jik876/hifi-gan

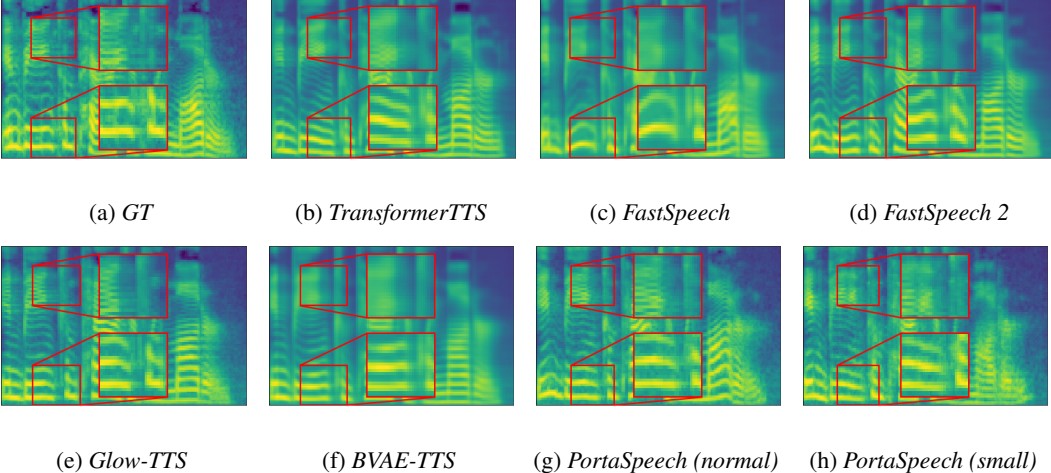

| (a) *GT* | (b) *TransformerTTS* | (c) *FastSpeech* | (d) *FastSpeech 2* |

| (e) *Glow-TTS* | (f) *BVAE-TTS* | (g) *PortaSpeech (normal)* | (h) *PortaSpeech (small)* |

Figure 3: Visualizations of the ground-truth and generated mel-spectrograms by different TTS models. The corresponding text is "*In being comparatively modern*".

phenomenon inspires us to apply VAE-based mel-spectrogram decoder (variational generator) to our lightweight TTS model. 2) Compared with flow-based models, VAE-based model has poorer audio quality upper bound according to MOS-Q, which motivates us to make up for shortcomings of VAE by introducing a flow-based post-net to refine the mel-spectrograms generated by VAE.

### 4.3 Performance

Table 2: The audio performance (MOS-Q and MOS-P), inference latency, peak memory (Peak Mem.) and number of model parameters (#Params.) comparisons. The evaluation is conducted on a server with 1 NVIDIA 2080Ti GPU and batch size 1. The mel-spectrograms are converted to waveforms using Hifi-GAN (V1) [11]. RTF denotes the real-time factor, that the seconds required for the system (together with Hifi-GAN vocoder) to synthesize one-second audio.

| Method | MOS-P | MOS-Q | RTF | Peak Mem. | #Params. |
|---|---|---|---|---|---|
| *GT* | $4.52 \pm 0.07$ | $4.41 \pm 0.06$ | / | / | / |
| *GT (voc.)* | $4.48 \pm 0.08$ | $4.15 \pm 0.07$ | / | / | / |
| *Tacotron 2 [29]* | $3.85 \pm 0.07$ | $3.80 \pm 0.08$ | 0.115 | 61.78MB | 28.2M |
| *TransformerTTS [15]* | $3.87 \pm 0.06$ | $3.82 \pm 0.07$ | 0.955 | 118.66MB | 24.2M |
| *FastSpeech [25]* | $3.63 \pm 0.08$ | $3.72 \pm 0.08$ | 0.0198 | 115.2MB | 23.5M |
| *FastSpeech 2 [24]* | $3.72 \pm 0.07$ | $3.83 \pm 0.06$ | 0.0200 | 124.8MB | 27.0M |
| *Glow-TTS [8]* | $3.61 \pm 0.07$ | $3.88 \pm 0.08$ | 0.0196 | 116.4MB | 28.6M |
| *BVAE-TTS [14]* | $3.80 \pm 0.06$ | $3.72 \pm 0.06$ | **0.0169** | 90.1MB | 12.0M |
| *PortaSpeech (normal)* | **$3.89 \pm 0.06$** | **$3.92 \pm 0.06$** | 0.0216 | 83.6MB | 21.8M |
| *PortaSpeech (small)* | $3.82 \pm 0.06$ | $3.86 \pm 0.06$ | 0.0208 | **39.3MB** | **6.7M** |

We compare the quality of generated audio samples, inference latency, model size[9] and memory footprint[10] of our PortaSpeech (normal and small model size) with other systems, including 1) *GT*, the ground truth audio; 2) *GT (Mel + HiFi-GAN)*, where we first convert the ground truth audio into mel-spectrograms, and then convert the mel-spectrograms back to audio using HiFi-GAN; 3) *Tacotron 2 [29]*; 4) *Transformer TTS [15]*; 5) *FastSpeech [25]*; 6) *FastSpeech 2 [24]*; 7) *Glow-TTS [8]* and 8) *BVAE-TTS [14]*[11]. The results are shown in Table 2. We have the following observations:

---

[9]The model parameters do not include the encoder of VAE in BVAE-TTS and PortaSpeech.

[10]We profile the peak GPU memory using *MemReporter* in *pytorch_memlab* (https://github.com/Stonesjtu/pytorch_memlab) and find the maximum "*active_bytes*" as the peak memory during inference.

[11]We fail in reproducing the performance of BVAE-TTS reported in the original paper, so we use hard text-to-speech alignment in their model and obtain reasonable results.

- For *audio quality*, PortaSpeech (normal) outperforms previous TTS models in both audio quality (MOS-Q) and prosody (MOS-P), and only has slight performance degradation when reducing the model size, which shows the superiority of our proposed method.

- For *model size* and *memory footprint*, PortaSpeech (small) has the smallest model size and memory footprint. Compared with FastSpeech 2, PortaSpeech (small) achieves 4x model size and 3x memory footprint compression ratios.

- For *inference speed*, PortaSpeech (small) speeds up the end-to-end speech generation by 5.5x and 45.9x compared with Tacotron 2 and TransformerTTS and achieves similar RTF with other NAR-TTS models.

Besides, we conduct some experiments on the multi-speaker dataset and draw similar conclusions (see Appendix C). We also conduct robustness evaluation on both single-speaker and multi-speaker dataset in Appendix D and find that PortaSpeech achieves comparable robustness performance with state-of-the-art NAR-TTS models.

## 4.4 Visualizations

We then visualize the mel-spectrograms generated by the above systems in Figure 3. We can see that PortaSpeech can generate mel-spectrograms with rich details in frequency bins between two adjacent harmonics, unvoiced frames and high-frequency parts, which results in natural sounds. Besides, we visualize the diverse mel-spectrograms generated by PortaSpeech in Appendix F. In conclusion, our experiments demonstrate that PortaSpeech achieves the goals described in Section 1 (*fast*, *lightweight*, *high-quality*, *expressive* and *diverse*).

## 4.5 Ablation Studies

We conduct ablation studies to demonstrate the effectiveness of designs in PortaSpeech, including the enhanced prior, our post-net and the mixture alignment. We put more analyses on the grouped parameter sharing mechanism in Appendix G. We conduct CMOS evaluation for these ablation studies. The results are shown in Table 3.

**Enhanced Prior** To demonstrate the effectiveness of enhanced normalizing flow-based prior, we compare our models with those with simple Gaussian prior as the original VAE. The results are shown in row 2 in Table 3. We can see that CMOS-P drops when removing the enhanced prior, indicating that the enhanced prior can improve the prosody. Since the prosody is mainly modeled by VAE, compared with simple Gaussian prior, the enhanced prior has weaker assumptions and restrictions on the shape of the VAE prior distribution.

Table 3: Audio prosody and quality comparisons for ablation study. *MA* denotes mixture alignment in the linguistic encoder; *PN* denotes the flow-based post-net; *EP* denotes the enhanced prior in the variational generator; *Conv* denotes the convolutional post-net used in Tacotron 2 [29].

| Setting | normal | | small | |
|---|---|---|---|---|
| | CMOS-P | CMOS-Q | CMOS-P | CMOS-Q |
| *PortaSpeech* | 0.000 | 0.000 | 0.000 | 0.000 |
| - *EP* | -0.194 | -0.014 | -0.212 | -0.098 |
| - *PN* | -0.012 | -0.458 | -0.007 | -0.162 |
| - *PN + Conv* | -0.010 | -0.441 | -0.005 | -0.148 |
| - *MA* | -0.241 | -0.127 | -0.312 | -0.157 |

**Post-Net** To demonstrate the effectiveness and necessity of flow-based post-net, we compare PortaSpeech with that without the post-net and that with convolutional post-net, which is widely used in previous TTS models, such as Tacotron 2 [29]. The results are shown in row 3 and row 4 in Table 3. From row 3, it can be seen that CMOS-Q drops significantly when removing our post-net, demonstrating that our post-net can improve the audio quality of the generated mel-spectrograms. From row 4, we can see that our flow-based post-net outperforms the commonly used convolutional post-net.

Table 4: Average absolute duration error comparisons in word and sentence level on test set for PortaSpeech (small).

| Settings | Word (ms) | Sentences (s) |
|---|---|---|
| *w/ MA* | 96.3 | 1.40 |
| *w/o MA* | 136.7 | 1.84 |

**Mixture Alignment**   To demonstrate the effectiveness of mixture alignment, we replace the mixture alignment in the linguistic encoder with the phoneme-level hard alignment proposed in Fast-Speech [25]. The results are shown in row 5 in Table 3. We can see that PortaSpeech with mixture alignment outperforms that with phoneme-level hard alignment in terms of both CMOS-P and CMOS-Q. These results demonstrate that 1) mixture alignment can improve the prosody, which may benefit from more accurate duration extraction and prediction; 2) mixture alignment can also improve the generated voice quality since the soft alignment helps the end-to-end model optimization. Then we calculate the average absolute duration error in word and sentence level on the test set for PortaSpeech (small) with and without mixture alignment. The results are shown in Table 4. It can be seen that the linguistic encoder with mixture alignment predicts more accurate duration, also demonstrating the effectiveness of the mixture alignment. We visualize the attention alignments generated by our linguistic encoder in Appendix A.1, showing that PortaSpeech can create reasonable alignments which is close to the diagonal.

## 5   Conclusion

In this paper, we proposed PortaSpeech, a portable and high-quality generative text-to-speech model. PortaSpeech uses a variational generator with an enhanced prior followed by a flow-based post-net with grouped parameter sharing mechanism as the main model architecture. We also proposed a new linguistic encoder with mixture alignment to improve the prosody and reduce the dependence on the hard fine-grained alignment, which combines the hard word-level and soft phoneme-level alignments. Our experimental results show that PortaSpeech outperforms other TTS models in voice quality and prosody and shows only a slight performance degradation when reducing the model size. We also conduct comprehensive ablation studies to verify the effectiveness of each component in PortaSpeech. However, to take advantage of the merits of VAE and normalizing flow, we sacrifice at the cost of more complicated model designs than previous NAR-TTS models: the overall architecture, which cascades linguistic encoder, VAE and post-net, is somewhat complicated. In the future, we will verify the effectiveness of PortaSpeech on multi-speaker and multilingual scenarios. We will also try to tap its potential on other tasks, such as voice conversion and end-to-end text-to-waveform generation.

## Acknowledgments

This work was supported in part by the National Key R&D Program of China under Grant No.2020YFC0832505, National Natural Science Foundation of China under Grant No.61836002, No.62072397, Zhejiang Natural Science Foundation under Grant LR19F020006 and Baidu Scholarship Program.

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
