# OpenReview forum: "PortaSpeech: Portable and High-Quality Generative Text-to-Speech"
_NeurIPS.cc/2021/Conference — NeurIPS 2021 Poster_

### Official Review · Reviewer_t4mP · 2021-07-15

**Rating:** 7
**Confidence:** 5

**Summary:**

To overcome the shortcomings that autoregressive text-to-speech (AR TTS) models have (i.e. high inference latency, low robustness), non-AR TTS models such as Glow-TTS and BVAE-TTS have recently been proposed, which show competitive performance compared to AR TTS models.

In this paper, a non-autoregressive (non-AR) TTS model called PortaSpseech is proposed, which effectively reduces the number of its parameters by combining VAE and normalizing flow, while showing better performance than Glow-TTS and BVAE-TTS. First, by conducting preliminary analyses on VAE and Flow, it shows which model has which advantages over the other: VAE is good at capturing prosody even with small size and Flow is good at reconstructing the frequency bin-wise details. Then, to fully utilize the advantage of each method, this paper proposes novel ways to combine the models. To help model capture prosody, lightweight VAE equipped with enhanced prior is used in decoder of the main model, and normalizing flow with grouped parameter sharing mechanism is used as a post-net, which is helpful to improve fidelity while maintaining a small number of parameters. In addition to that, it also proposes a linguistic encoder with a mixture alignment mechanism, which is effective in learning prosody compared to the phoneme-level hard alignment.

Through various experiments, this paper supports its claim that PortaSpeech can generate high-quality speech with a small number of parameters. In preliminary analyses, by measuring MOS in terms of prosody and quality separately, it shows the characteristic that each VAE and Flow model has. Then, it compares PortaSpeech with previous non-AR TTS models and shows its superiority. Plus, ablation studies also show that each architectural component is really effective to improve the performance of PortaSpeech.

**Limitations And Societal Impact:**

As mentioned in the main review, I think it would have more analytic experiments.

Related to the preliminary study, when comparing VAE and Flow architectures, since the architectures share the same encoder and length regulator, I wonder about the performance gap of capturing prosody beyond the durations.  I think it could be better if there is an analytic experiment showing the reason how VAE can help model learn prosody.

As mentioned in Section 2, there have been many models trying to reduce the size of TTS models, so it could be better if PortaSpeech is compared with the models such as SpeedySpeech and LightSpeech. (although I am aware that the models approach TTS in different ways such as teacher-student model or NAS)

I hope the idea of combining VAE and flow makes it possible to develop an end-to-end TTS model such as VITS in "Conditional Variational Autoencoder with Adversarial Learning for End-to-End Text-to-Speech, Kim et al".


**Main Review:**

This paper gives useful guidance about, in which part, each of the generative model frameworks, VAE and Flow, is helpful for TTS models to have better performance. Especially, speaking of prosody, since it is not a general concept applied to other general generative models, I think it is quite remarkable and can give a good intuition for other researchers to develop other TTS models.

However, I think it would be better if there were deeper analyses because it seems that most claims are supported by comparing MOS-P/Q. Personally, it sounds sensible that Flow is helpful in generating high-quality speech, but it sounds a little weak that VAE has an advantage in capturing prosody.

In terms of clarity, when I read this paper, I am a little confused about the follows:
* For me, the word 'inter-word alignment' makes me think there is an attention mechanism between words as in Machine Translation.
* The paper says that Lightweghit VAE is used in the decoder, but I cannot figure out how the VAE is lightweight. I think it could be better if the number of parameters of each module is separately written in the appendix.
* I am a little confused when I am reading about enhanced prior. When I see figure 3-(c), the dotted line means residual connection? What original prior is used? How the prior is enhanced? I think the explanation about it could be added, and I suppose that there would be a reference of a paper "Improved Variational Inference with Inverse Autoregressive Flow, kingma et al."
* In section 3.4, are the coarse-grained mel-spectrograms just the mel-spectrograms before post-net? I understand in that way, but the word 'coarse-grained' makes me think that it is a down-sampled one or something.

Overall, I think this paper gives a valuable guide to other researchers to develop TTS models using the VAE and Flow generative models and I think this paper could be better if there were much more analytic experiments.

**Time Spent Reviewing:**

5

---

> ### Author Response · Authors · 2021-08-10
> **Response to Reviewer t4mP**
>
> Thanks for your comments!
>
> **[Deeper Analyses on VAE and Flow for NAR-TTS]**
> In image generation tasks, VAE is good at capturing the overall image structure information (low-frequency parts) while discarding small sharp textures/details (high-frequency parts). Similarly, in mel-spectrograms, low-frequency parts correspond to the shape of harmonics, which determines the pitch and prosody of speech. Thus we can intuitively infer that VAE is good at modeling the prosody while not good at modeling the details in speech.  We will add these analyses to the new version of the paper.
>
> **[About "Inter-Word Alignment"]**
> Thanks for your advice! We will change *"inter-word alignment"* to *"word-level alignment"* and change *"intra-word alignment"* to *"phoneme-level alignment"* in the new version of the paper.
>
> **[Number of Parameters of Each Module]**
> We list the number of parameters of each module in the following table. We can see that the parameters of Variational Generator (VG) are much fewer than that of Post-Net in PortaSpeech (normal) and PortaSpeech (small). We will add these details to the new version of the paper.
>
> | Module              | Normal    | Small     |
> | ------------------- | --------- | --------- |
> | Linguistic Encoder  | 7.2M      | 2.0M      |
> | Duration predictor  | 0.3M      | 0.2M      |
> | Post-Net            | 10.8M     | 3.6M      |
> | Decoder in VG       | 2.5M      | 0.6M      |
> | VP-Flow in VG       | 1.0M      | 0.3M      |
> | Total               | 21.8M     | 6.7M      |
>
> **[About the Enhanced Prior]**
> The dotted line means the procedure that is only used in training. We describe more details of the enhanced prior:
>
> - In training, the posterior distribution ($\mu_q$, $\sigma_q$) is encoded by the encoder of the variational generator. Then $z_q$ is sampled from the posterior distribution using reparameterization and is passed to the decoder of the variational generator (the right dotted line). In the meanwhile, the posterior distribution is fed into the VP-Flow to convert it to a standard normal distribution (the middle dotted line).
> - In inference, VP-Flow converts a sample in the standard normal distribution into a sample $z_p$ in the prior distribution of the variational generator and we pass the $z_p$ to the decoder of the variational generator. In this way, the prior of variational generator is no longer a standard normal distribution.
>
> We will add these descriptions to the new version of the paper. We will also add the missing reference: *"Improved Variational Inference with Inverse Autoregressive Flow, kingma et al."* to the new version of the paper.
>
> **[About the "Coarse-grained"]**
> Yes, the coarse-grained mel-spectrograms is the mel-spectrograms before post-net. We will clarify this in the new version of the paper.
>
> **[Comparisons with Lightweight TTS Models]**
> We conduct CMOS evaluation to compare PortaSpeech (small, 6.7M) with SpeedySpeech (4.3M) on LJSpeech. The result shows that PortaSpeech achieves better prosody (+0.317 CMOS-P) and audio quality (+0.359 CMOS-Q) than SpeedySpeech. We further reduce the model size of PortaSpeech to match that of SpeedySpeech to 4.25M and find that PortaSpeech still outperforms SpeedySpeech (+0.254 CMOS-P and +0.198 CMOS-Q). We will add these comparisons to the new version of the paper. As for LightSpeech, we do not find the official implementation.
>
> **[End-to-End TTS Model]**
> We've designed an end-to-end text-to-waveform model based on the PortaSpeech architecture: the output of VAE decoder is waveform instead of mel-spectrograms and is trained with multi-scale STFT losses as used in Parallel WaveGAN [1]. Then the output waveform of VAE is passed to the glow-based post-net as the condition. The post-net finally generates the refined waveforms. However, due to the time limitation of the rebuttal period, our designed text-to-waveform model has not achieved the SOTA performance. So it will be regarded as our future work.
>
>
> ***[References]***
> *[1] Yamamoto, Ryuichi, Eunwoo Song, and Jae-Min Kim. "Parallel WaveGAN: A fast waveform generation model based on generative adversarial networks with multi-resolution spectrogram." ICASSP 2020-2020 IEEE International Conference on Acoustics, Speech and Signal Processing (ICASSP). IEEE, 2020.*

---

> > ### Comment · Reviewer_t4mP · 2021-08-15
> > **Response to the author's response**
> >
> > Thank you for your careful response to my opinion. After I read your response, I come to think that your paper can be much better and can give much intuition to other Text-to-Speech researchers. I hope your plan will be reflected to your revised version, especially your intuition about how VAE can help model capture prosody. I will raise my score from 6 to 7.

---

### Official Review · Reviewer_e88V · 2021-07-17

**Rating:** 6
**Confidence:** 2

**Summary:**

This paper summarizes a text-to-speech system which uses a VAE with a normalizing flow-based prior to generate spectrograms, and then uses a GAN model to convert the synthesized spectrogram to a waveform.


**Ethics Review Area:**

["I don’t know"]

**Limitations And Societal Impact:**

The authors do not discuss this.

**Main Review:**

Results indicate a very slight improvement in terms of quantitative metrics. In terms of qualitative synthesis quality to me the proposed model does not sound perceptibly better.

The `small' version of the proposed model seems to reduce the number of parameters and the memory usage, and this might be useful in edge device applications. I am not exactly sure if the existing TTS models do not have more efficient variants though.

Overall, the paper seems to propose a valid model, and although not ground breaking, the model seems to be doing a decent job in terms of generation quality, and being memory efficient.

**Time Spent Reviewing:**

1

---

> ### Author Response · Authors · 2021-08-10
> **Response to Reviewer e88V**
>
> Thanks for your comments!
>
> **[About the Qualitative Synthesis Quality]**
> Previous methods suffer from the over-smoothing/blurry problem, which hurts the mel-spectrogram details and the high-frequency parts, thus resulting in unnatural sounds, and it is consistent with the findings we observed from visualizations in Figure 3. As for the prosody, we add more objective evaluation on the pitch contour to verify if PortaSpeech achieves better prosody: we compare the moments of the pitch in ground-truth and synthesized audio, including standard deviation ($\sigma$), skewness ($\gamma$) and kurtosis ($\mathcal{K}$) on the test set, following FastSpeech 2 [2]. The results are shown in the following table. We can see that the moments of generated audio of PortaSpeech are closer to the ground-truth audio, demonstrating that PortaSpeech can generate speech with more natural pitch contour. We will add these experimental analyses to the new version of the paper.
>
> | Method               | $\sigma$ |$\gamma$  | $\mathcal{K}$|
> | -------------------- | ---------|--------- | --------- |
> | GT                   | 54.8     |0.842     | 0.964    |
> | Tacotron 2           | 47.2     |0.935     | 1.218    |
> | FastSpeech           | 46.2     |0.711     | 0.215     |
> | FastSpeech 2         | 50.2     |0.871     | 1.121     |
> | PortaSpeech (normal) | **52.3** |**0.853** | **1.011** |
> | PortaSpeech (small)  | 51.8     |0.869     | 1.112     |
>
> **[About Efficient Variants in Existing TTS Models]**
> From Table 1 and Table 2, we can find that small version PortaSpeech outperforms VAE and Glow-based models in terms of MOS-P and MOS-Q. For FastSpeech, FastSpeech 2 and BVAE-TTS with normal size, we can see that PortaSpeech (small) outperforms all of them. When reducing their number of parameters, their performance will be further dropped. These comparisons with mainstream TTS models can demonstrate the parameter efficiency of our PortaSpeech.
>
> **[About Limitations And Societal Impact]**
> We listed the limitations in Section 5 in the paper and societal impact in Appendix F in the originally submitted supplementary material.

---

> > ### Comment · Reviewer_e88V · 2021-08-18
> > **Response to the rebuttal**
> >
> > Thanks for your answers. I am suggesting to accept the paper.

---

### Official Review · Reviewer_LAM4 · 2021-07-17

**Rating:** 7
**Confidence:** 3

**Summary:**

This paper combines the high quality generation abilities of a flow network with the prosodic variation of a VAE, successfully avoiding the drawbacks of each architecture, to produce a single-speaker TTS model that can be scaled down significantly while still achieving close to state-of-the-art results.

**Limitations And Societal Impact:**

The authors mention the potential negative societal impact, but do not offer any suggestions.

**Main Review:**

The idea is well motivated and the paper is well written. My biased subjective opinion is that there's nothing groundbreaking so this would probably not be one of the top papers but it is solid.


3.2 Variational Generator with Enhanced Prior

It seems to me that instead of treating the complex distribution as the prior, this setup should be equivalent to continue using a gaussian prior, but adding the reverse flow and flow to the VAE encoder and decoder respectively?

Which makes me think that a gaussian prior with a sufficiently large encoder/decoder should still work just as well. The ablation experiment in the paper only removed the flow, but did not increase the size of the encoder/decoder by the size of the flow, so it is not exactly a fair comparison. It is very possible that the parameter coupling happening in the flow is key to making it work well though, which would be an interesting conclusion.


3.3 Flow-based Post-Net

This is outside of the scope of this work, but we know flow-based methods can generate waveforms directly ala WaveGlow. Have you considered doing so instead of generating mel spectrograms, which should further simplify the mobile TTS story?


4.1 Experimental Setup

In my opinion, LJSpeech is not a great dataset. It is single speaker and does not contain a lot of prosody variation. It would have been better to see results on something like LibriTTS.


4.3 Performance

I find the comparison of pitch accuracy a bit strange. For models with VAE it would make sense to use reference posterior latents and check that pitch contours match. But otherwise, it's expected that given a piece of text there can be many valid ways to vocalize it and so a higher pitch accuracy isn't necessarily always a positive result for a TTS model.


Mixture Alignment

Can the mixture alignment be used to specify absolute durations for individual words at inference time? What about for silences? If it gives this capability then this can be an important feature and is superior to the alignment in FastSpeech which can only do relative scaling.


# Originality
VAE-based and flow-based non-autoregressive TTS models are not new. I imagine there have been plenty of experiments combining the two as well. The use of a volume-preserving flow inside the VAE is novel. The ability to reduce the model size while maintaining quality with parameter sharing in the flow might be novel.


# Quality
The submission is high quality and includes relevant experiments. Some experiments on a tougher dataset would make it a better submission.


# Clarity
The submission is clear and well organized.


# Significance
The results can be a good step towards high quality mobile TTS, though it is hard to say how much traction this can gain compared to approaches like sparsifying/compressing more well known models. If the flow could be altered to generate waveforms directly while matching the quality of FastSpeech2+Parallel WaveGAN, this could immediately become a top paper in TTS.

**Time Spent Reviewing:**

2

---

> ### Author Response · Authors · 2021-08-10
> **Response to Reviewer LAM4**
>
> Thanks for your comments!
>
> **[About the Variational Generator with Enhanced Prior]**
> We further conduct ablation studies to compare PortaSpeech (normal) with the model (A) without enhanced prior but with the same model parameters (by increasing the size of the decoder). The results show that PortaSpeech still achieves better prosody (+0.188 CMOS-P) compared with this model (A). We think it is possible that the parameter coupling mechanism in flow can make the model more compact while keeping the flexibility of the prior distribution.
>
> **[About Flow-based Post-Net for Text-to-Waveform Generation]**
> Yes, we've designed an end-to-end text-to-waveform model based on PortaSpeech architecture: the output of VAE decoder is waveform instead of mel-spectrograms and is trained with multi-scale STFT losses as used in Parallel WaveGAN [1]. Then the output waveform of VAE is passed to the glow-based post-net as the condition. The post-net finally generates the refined waveforms. However, due to the time limitation of the rebuttal period, our designed text-to-waveform model has not achieved the SOTA performance. So it will be regarded as our future work.
>
> **[About Multi-Speaker Datasets]**
> We conduct the MOS evaluation on the multi-speaker dataset: LibriTTS. The results are shown in the following table (we use a pre-trained Parallel WaveGAN [1] for LibriTTS as the vocoder). We can draw similar conclusions as that on LJSpeech that PortaSpeech can achieve good prosody and audio quality in terms of MOS-P and MOS-Q, even in more complicated (multi-speaker) scenarios. We will add these experimental results to the new version of the paper.
>
> | Method               | MOS-P         | MOS-Q         |
> | -------------------- | ------------- | ------------- |
> | GT                   | 4.24±0.08     | 4.36±0.09     |
> | GT (vocoder)         | 4.21±0.09     | 4.01±0.10     |
> | Tacotron 2           | 3.81±0.10     | 3.71±0.11     |
> | TransformerTTS       | 3.79±0.09     | 3.72±0.12     |
> | FastSpeech           | 3.59±0.11     | 3.61±0.14     |
> | FastSpeech 2         | 3.64±0.11     | 3.70±0.11     |
> | Glow-TTS             | 3.76±0.15     | 3.78±0.10     |
> | PortaSpeech (normal) | **3.84±0.13** | **3.83±0.13** |
> | PortaSpeech (small)  | 3.80±0.12     | 3.81±0.11     |
>
> **[About the Pitch Accuracy]**
> To determine whether PortaSpeech can generate more natural prosody, we compare the moments of the pitch contour in ground-truth and synthesized audio, including standard deviation ($\sigma$), skewness ($\gamma$) and kurtosis ($\mathcal{K}$) on the test set, following FastSpeech 2 [2]. The results are shown in the following table. We can see that the moments of generated audio of PortaSpeech are closer to the ground-truth audio, demonstrating that PortaSpeech can generate speech with more natural pitch contours. We will add these experimental analyses to the new version of the paper.
>
> | Method               | $\sigma$ |$\gamma$  | $\mathcal{K}$|
> | -------------------- | ---------|--------- | --------- |
> | GT                   | 54.8     |0.842     | 0.964    |
> | Tacotron 2           | 47.2     |0.935     | 1.218    |
> | FastSpeech           | 46.2     |0.711     | 0.215     |
> | FastSpeech 2         | 50.2     |0.871     | 1.121     |
> | PortaSpeech (normal) | **52.3** |**0.853** | **1.011** |
> | PortaSpeech (small)  | 51.8     |0.869     | 1.112     |
>
> **[About the Duration in Mixture Alignment]**
> Since we use hard inter-word alignment in PortaSpeech, absolute durations for individual words can be specified at inference time. As for silences, we can add a word boundary symbol as an extra special word such as "<SIL>" between two words in training. In this way, we can adjust the duration of silences via modifying the duration of the special word "<SIL>". So PortaSpeech can surely do relative scaling and we will add these experimental analyses to the new version of the paper.
>
> ***[References]***
> *[1] Yamamoto, Ryuichi, Eunwoo Song, and Jae-Min Kim. "Parallel WaveGAN: A fast waveform generation model based on generative adversarial networks with multi-resolution spectrogram." ICASSP 2020-2020 IEEE International Conference on Acoustics, Speech and Signal Processing (ICASSP). IEEE, 2020.*
> *[2] Ren, Yi, et al. "FastSpeech 2: Fast and High-Quality End-to-End Text to Speech." International Conference on Learning Representations. 2020.*

---

> > ### Comment · Reviewer_LAM4 · 2021-08-16
> > **Response**
> >
> > Thank you. I will bump up my rating from 6 to 7 for the new experimental results on LibriTTS and comparisons to SpeedySpeech.

---

### Official Review · Reviewer_zUv7 · 2021-07-17

**Rating:** 6
**Confidence:** 4

**Summary:**

The authors propose PortaSpeech, a non-autoregressive model that can sythesize high-quality speech with reduced model size. Based on the experiment results, PortaSpeech slightly outperforms counterpart models in speech quality and shows only a slight performance degradtaion when reducing the model size.

**Limitations And Societal Impact:**

The authors have adequately addressed the limitations and potential negative societal impact of their work.

**Main Review:**

Strength:
The proposed mixture alignment is novel.
PortaSpeech has good MOS ratings even with reduced model size.

Weekness:
The comparsion with [1] is missing since [1] also leverages a variational mel-spectragram encoder.
The comparsion of proposed mixture alignment with gaussian alignment [2][3][4] is missing
The model requires external aligner for training while other NAR models such as EATS [2], Glow-TTS, EfficientTTS [4] can be trained end-to-end.
The robustness evalutaion is missing.

[1] Isaac Elias, et al. Parallel Tacotron: Non-Autoregressive and Controllable TTS. ICASSP 2021

[2] Jeff Donahue, et al. End-to-end adversarial text-to-speech. In ICLR, 2021.

[3] Jonathan Shen, et al. Non-attentive Tacotron: Robust and controllable neural TTS synthesis including unsupervised duration modeling. arXiv:2010.04301, 2020.

[4] Chenfeng Miao, et al. EfficientTTS: An efficient and high-quality text-to-speech architecture. ICML 2021


**Time Spent Reviewing:**

8 hours

---

> ### Author Response · Authors · 2021-08-10
> **Response to Reviewer zUv7**
>
> Thanks for your comments!
>
> **[Comparison with Parallel Tacotron]**
> Parallel Tacotron [1] and PortaSpeech both capture the variance in speech (e.g., prosody) using VAE. However, there are several differences between them: 1) To model the word-level prosody better, Parallel Tacotron uses attention to aggregate the word-level latent representation in VAE, while we directly downsample the mel-spectrogram frames and use mixture alignment in linguistic encoder; 2) we introduce a flow-based component to enhance the prior in VAE to better model the variance in speech; and 3) Parallel Tacotron applies multi-head LConv blocks to improve the generated speech in an autoregressive way, while we use a fully parallel mel-spectrogram decoder and a flow-based post-net to generate high-quality speech. We will add this comparison to the new version of the paper. However, there is no public implementation of Parallel Tacotron, and we fail to reimplement the performance reported in its original paper. So we cannot conduct the quantitative performance comparison with Parallel Tacotron.
>
> **[Comparison with Gaussian Alignment-Based Methods]**
> EATS [2], Non-attentive Tacotron [3] and EfficientTTS [4] have not released their codes officially. As for EfficientTTS, we find an unofficial implementation (https://github.com/liusongxiang/efficient_tts) and compare it with our PortaSpeech. We conduct P-CMOS and Q-CMOS evaluations and the results show that PortaSpeech outperforms EfficientTTS in both prosody (+0.295 CMOS-P) and quality (+0.352 CMOS-Q).  We will add these results to the new version of the paper.
>
> **[Comparison with End-to-End Aligner]**
> Yes, PortaSpeech needs an external aligner to obtain the word-level alignment. Although the training pipeline is more complicated than those with end-to-end aligners, we find that the external aligner in our method is more robust in multi-speaker scenarios. We demonstrate this in the table **(Results for models trained on LibriTTS)** in the **robustness evaluation** part (PortaSpeech v.s. Glow-TTS [5]). We will add this comparison to the new version of the paper.
>
> **[Robustness Evaluation]**
> We conduct the robustness evaluation on LJSpeech and LibriTTS datasets. We select 50 sentences that are particularly hard for TTS systems following FastSpeech [6]. The results are shown in the following tables. We can see that PortaSpeech achieves comparable robustness performance with state-of-the-art non-autoregressive TTS models. We will add this evaluation to the new version of the paper.
>
> **Results for models trained on LJSpeech**
>
> *(Errors in sentences include repeats, skips and wrong pronunciations.)*
>
> | Method               | Repeats | Skips | Error Sentences |
> | -------------------- | ------- | ----- | --------------- |
> | Tacotron 2           | 4       | 5     | 7              |
> | TransformerTTS       | 7       | 7     | 9              |
> | FastSpeech           | 0       | 1     | 1               |
> | FastSpeech 2         | 0       | 1     | 1               |
> | Glow-TTS             | 0       | 2     | 2               |
> | PortaSpeech (normal) | 1       | 0     | 1               |
> | PortaSpeech (small)  | 1       | 1     | 1               |
>
> **Results for models trained on LibriTTS**
>
> *(We randomly choose one speaker for each sentence.)*
>
> | Method               | Repeats | Skips | Error Sentences |
> | -------------------- | ------- | ----- | --------------- |
> | Tacotron 2           | 6       | 7     | 12              |
> | TransformerTTS       | 10      | 12    | 15              |
> | FastSpeech           | 2       | 1     | 2               |
> | FastSpeech 2         | 2       | 1     | 2               |
> | Glow-TTS             | 5       | 4     | 8               |
> | PortaSpeech (normal) | 1       | 2     | 2               |
> | PortaSpeech (small)  | 2       | 2     | 2               |
>
> ***[References]***
> *[1] Isaac Elias, et al. Parallel Tacotron: Non-Autoregressive and Controllable TTS. ICASSP 2021*
> *[2] Jeff Donahue, et al. End-to-end adversarial text-to-speech. In ICLR, 2021.*
> *[3] Jonathan Shen, et al. Non-attentive Tacotron: Robust and controllable neural TTS synthesis including unsupervised duration modeling. arXiv:2010.04301, 2020.*
> *[4] Chenfeng Miao, et al. EfficientTTS: An efficient and high-quality text-to-speech architecture. ICML 2021*
> *[5] Kim J, Kim S, Kong J, et al. Glow-TTS: A Generative Flow for Text-to-Speech via Monotonic Alignment Search[J]. Advances in Neural Information Processing Systems, 2020, 33.*
> *[6] Ren, Yi, et al. "FastSpeech: fast, robust and controllable text to speech." Proceedings of the 33rd International Conference on Neural Information Processing Systems. 2019.*

---

> > ### Comment · Reviewer_t4mP · 2021-08-15
> > **Implementation for other TTS models.**
> >
> > As one of the text-to-speech researchers, I am really sorry that most of the papers do not release their source code, which makes us really difficult to fairly compare the TTS models. However, as it is written in the repository, I think that the unofficial implementation of Efficient-TTS is too deficient to be used :(

---

> > > ### Comment · Reviewer_zUv7 · 2021-08-21
> > > **The unofficial implementation implementation of Efficient-TTS is too deficient to be used**
> > >
> > > Yes, I agree that the unofficial implementation implementation of Efficient-TTS is too deficient to be used. The P-CMOS and Q-CMOS evaluations are not meaningful.

---

### Decision · Program_Chairs · 2021-09-27

**Decision:**

Accept (Poster)

**Comment:**

The paper presents an interesting blend of VAE and flow based approach for TTS. The reviewers raised several points about the comparisons -- including that some of the baselines are possibly not as good as the original work, since original implementations were not released and third party implementations had to be used for comparison. The authors address a lot of these concerns in the discussions and added analyses and clarification that I hope will make it to the final submission as they strengthen the presentation significantly. Thanks, to the reviewers for their constructive suggestions.